# Genomic Characterization of Carbapenem-Resistant Bacteria from Beef Cattle Feedlots

**DOI:** 10.3390/antibiotics12060960

**Published:** 2023-05-25

**Authors:** Sani-e-Zehra Zaidi, Rahat Zaheer, Krysty Thomas, Sujeema Abeysekara, Travis Haight, Luke Saville, Matthew Stuart-Edwards, Athanasios Zovoilis, Tim A. McAllister

**Affiliations:** 1Lethbridge Research and Development Centre, Agriculture and Agri-Food Canada, Lethbridge, AB T1J 4B1, Canada; saniezehra.zaidi@agr.gc.ca (S.-e.-Z.Z.); rahat.zaheer@agr.gc.ca (R.Z.); krysty.thomas@agr.gc.ca (K.T.); sujeema.abeysekara@agr.gc.ca (S.A.); 2Department of Chemistry and Biochemistry, University of Lethbridge, 4401 University Drive, Lethbridge, AB T1K 3M4, Canada; haight@uleth.ca (T.H.); luke.saville@uleth.ca (L.S.); m.stuartedwards@uleth.ca (M.S.-E.); athanasios.zovoilis@uleth.ca (A.Z.)

**Keywords:** carbapenem-resistant bacteria, antimicrobial resistance, beef production system, whole genome sequencing

## Abstract

Carbapenems are considered a last resort for the treatment of multi-drug-resistant bacterial infections in humans. In this study, we investigated the occurrence of carbapenem-resistant bacteria in feedlots in Alberta, Canada. The presumptive carbapenem-resistant isolates (*n* = 116) recovered after ertapenem enrichment were subjected to antimicrobial susceptibility testing against 12 different antibiotics, including four carbapenems. Of these, 72% of the isolates (*n* = 84) showed resistance to ertapenem, while 27% of the isolates (*n* = 31) were resistant to at least one other carbapenem, with all except one isolate being resistant to at least two other drug classes. Of these 31 isolates, 90% were carbapenemase positive, while a subset of 36 ertapenem-only resistant isolates were carbapenemase negative. The positive isolates belonged to three genera; *Pseudomonas*, *Acinetobacter*, and *Stenotrophomonas*, with the majority being *Pseudomonas aeruginosa* (*n* = 20) as identified by 16S rRNA gene sequencing. Whole genome sequencing identified intrinsic carbapenem resistance genes, including *blaOXA-50* and its variants (*P. aeruginosa*), *blaOXA-265* (*A. haemolyticus*), *blaOXA-648* (*A. lwoffii*), *blaOXA-278* (*A. junii*), and *blaL1* and *blaL2* (*S. maltophilia*). The acquired carbapenem resistance gene (*blaPST-2*) was identified in *P. saudiphocaensis* and *P. stutzeri*. In a comparative genomic analysis, clinical *P. aeruginosa* clustered separately from those recovered from bovine feces. In conclusion, despite the use of selective enrichment methods, finding carbapenem-resistant bacteria within a feedlot environment was a rarity.

## 1. Introduction

Carbapenems are β-lactam antibiotics that consist of a four-membered β-lactam ring fused with a secondary five-membered thiazolidine ring through the nitrogen and adjacent tetrahedral carbon atom. Unlike other β-lactams, carbapenems have two substitutions, at position one there is a substitution of sulfur for a carbon atom and at the fourth position of the thiazolidinic moiety, a carbon is substituted for a sulfone [1,2]. So far, four carbapenems, including ertapenem, meropenem, doripenem, and imipenem, have been approved for use in the US. These members differ in their side chains, influencing their antimicrobial activity. Carbapenems inhibit cell wall synthesis by preventing the formation of cross-linkages in peptidoglycan via binding to peptidoglycan binding protein (PBP), thus leading to cell lysis and death [3]. The ability of carbapenems to bind to diverse PBPs with high affinity and their stability against extended-spectrum β-lactamases (ESBLs) and AmpC β-lactamases account for their broad-spectrum activity against both Gram-negative and Gram-positive bacteria [2,4,5]. Ertapenem binds preferentially to PBPs 2 and 3 of *Escherichia coli* and has a low affinity for PBP 1a, 1b, 4, and 5. Imipenem binds with high affinity to PBP2, followed by PBP1a and 1b, but binds to PBP3 with low affinity. Meropenem possesses a high affinity for PBP 2, 3, and 4. Doripenem, similar to meropenem can bind to PBPs 2 and 3 of *P. aeruginosa* and also has affinity for PBP2 of *E. coli* [2]. The increased prevalence of resistance to penicillin, cephalosporins, fluoroquinolones and aminoglycosides has resulted in a significant increase in the clinical use of carbapenems [6].

Bacteria may circumvent carbapenem hydrolysis through intrinsic or acquired resistance mechanisms including the production of β-lactamases, overexpression of efflux pumps and mutations that alter the expression or/and function of porin proteins and PBPs [7,8,9]. The β-lactamases conferring resistance against carbapenem may belong to one of the Ambler classes [10] including class A (e.g., KPC), class B (e.g., VIM, NDM), and class D (e.g., OXA-48). Carbapenem resistance was first reported in opportunistic and environmental bacteria with intrinsic resistance [11]. Since then, carbapenem-resistant bacteria have been reported in the Asia-Pacific, India, Europe, and North and Latin America [12]. Resistant isolates have been recovered from a variety of sources including human clinics [11], livestock, including dairy cattle [13,14], beef cattle [15,16,17], and swine [18], sewage water [19], and wildlife [20,21].

Consequently, carbapenem resistance in Gram-negative bacteria has become a global problem with carbapenem-resistant Enterobacteriaceae (CRE), *Pseudomonas aeruginosa*, and *Acinetobacter baumannii* listed by the World Health Organization as being on the “ESKAPE” list of pathogens for which the control and development of new antimicrobials is urgently needed. Considering the importance of carbapenem in human health, the objective of this study was to investigate if carbapenem-resistant bacteria could be recovered from beef cattle feedlots through carbapenem enrichment. Initially, our focus species was *E. coli* as it serves as indicator bacteria in an antimicrobial resistance surveillance program within the One-health context. Later, recovered resistant isolates belonging to species other than *E. coli* were also included in downstream characterization using phenotypic and genotypic approaches. Furthermore, we investigated genomic relatedness among *Pseudomonas aeruginosa* isolates recovered in this study to previously published *P. aeruginosa* genomes through comparative analysis.

## 2. Results

### 2.1. Recovery of Carbapenem-Resistant Isolates and Species Identification

A total of 116 presumptive carbapenem-resistant isolates were recovered from bovine fecal and catch basin samples. Among these 8 were *E. coli*, 100 were *Pseudomonas* spp., 5 were *Acinetobacter* spp., 2 were *Ochrobactrum intermedium* (*Brucella intermedia*), and 1 was identified as *Stenotrophomonas maltophilia*.

### 2.2. Phenotypic Characterization

Figure 1 shows the antimicrobial resistance profiles of 116 isolates tested using antibiotic discs. Of all tested *E. coli* (*n* = 8) isolates, one isolate was resistant to ertapenem and tetracycline, while all others exhibited intermediate resistance to ertapenem. All *E. coli* isolates were negative for carbapenemase production (Figure 1).

All *Acinetobacter* spp. isolates (*n* = 5) were resistant to both ertapenem and meropenem and positive for carbapenemase production. One isolate of *Acinetobacter* spp. was also resistant to tetracycline and trimethoprim-sulfonamide. Tobramycin resistance was also identified in two *Acinetobacter* spp. isolates recovered from bovine feces (Figure 1).

For *Pseudomonas* spp., 7% of isolates were resistant to meropenem alone, 20% were resistant to both doripenem and meropenem, and 11% were resistant to all tested carbapenems. After carbapenems, *Pseudomonas* spp. were most commonly resistant to trimethoprim-sulfonamides (33%) followed by chloramphenicol (30%), tetracycline (23%), ceftazidime (8%), gentamicin (6%), tobramycin (6%), piperacillin (5%), and levofloxacin (5%) (Figure 1). A total of 56 *Pseudomonas* spp. were tested for carbapenemase production, and 38% of these isolates were confirmed positive.

A single *O. intermedium* isolate was resistant to ertapenem and meropenem, with both *O. intermedium* isolates being negative for carbapenemase production. The single *S. maltophilia* isolate identified in this study was resistant to all tested antibiotics and was positive for carbapenemase production (Figure 1 and Table 1).

### 2.3. Genomic Characterization

The illumina short read sequencing generated reads with an average sequencing coverage of 80% per isolate (Appendix A). Hybrid assembly using Illumina short and Flye-assembled long reads generated an average of six contigs per *P. aeruginosa* genome, with an average genome size of 6,581,091 bp and GC% content of 66.23%. The average genome sizes for *P. plecoglossicida* and *A. haemolyticus* were 5,836,945 bp (GC content = 62.24%) and 3,327,176 bp (GC content = 39.71%). The genome sizes of other *Pseudomonas* spp. including *P. mosselii*, *P. entomophila*, *P. putida*, *P. stutzeri*, and *P. saudiphocaensis* were 5,724,129 bp (GC content = 64.56%), 5,483,085 (GC content = 62.55%), 5,784,665 bp (GC content = 61.80%), 4,059,836 bp (GC content = 63.28%), and 3,671,120 bp (GC content = 61.14%), respectively. Genomes of *Acinetobacter lwoffii* and *A. junii* had 3,648,566 bp (GC content = 40.38%) and 3,109,155 bp (43.13%), while *O. intermedium* and *S. maltophilia* genomes were 4,881,352 bp (GC content = 57.53%) and 5,264,819 bp (GC content = 66.60%). The whole genome sequence data of the 42 bacterial isolates have been deposited in GenBank under BioProject PRJNA956966.

Sequencing of the ertapenem-resistant *E. coli* isolate (*n* = 1) revealed no antimicrobial resistance genes (ARG) associated with carbapenem resistance. However, ARGs associated with β-lactams (*blaEC*), cephalosporins (*blaCMY-2*), sulfonamide (*sul2*), aminoglycosides (*aph(3″)-Ib*, *aph(6)-ld*), tetracycline (*tetA*), and fluoroquinolones (*floR*) were identified (Table 1). All ARGs but *blaEC* were mapped to a conjugative plasmid (CP025245). The virulence genes associated with type II secretion system (T2SS), type III secretion system (T3SS), pili and fimbriae synthesis, adhesion, iron import, curli biogenesis, and enterobactin synthesis were also present in the *E. coli* genome. (Table 2).

A total of 34 *Pseudomonas* spp. were sequenced including *P. aeruginosa* (*n* = 20), *P. plecoglossicida* (*n* = 09), *P. entomophila* (*n* = 1), *P. mosselii* (*n* = 1), *P. putida* (*n* = 1), *P. saudiphocaensis* (*n* = 1), and *P. stutzeri* (*n* = 1). All *P. aeruginosa* isolates carried *blaOXA*-type carbapenemase (class D) gene [*blaOXA-50* (13/20, 65%), *blaOXA-486* (3/20, 15%), *blaOXA-494* (1/20, 5%), *blaOXA-902* (1/20, 5%), *blaOXA-648* (1/20, 5%)] (Table 1). Additionally, we identified ARGs associated with cephalosporin [*blaPDC-197* (9/20, 45%), *blaPDC-374* (3/20, 15%), *blaPDC-55* (3/20, 15%), *blaPDC-133* (1/20, 5%), *blaPDC-66* (1/20, 5%)], aminoglycoside [*aph(3′)-Iib* (18/20, 90%), *aph(3″)-Ia* (1/20, 5%), *aph(6)-Id* (1/20, 5%)], chloramphenicol [*catB7* (18/20, 90%)], fluoroquinolone [*crpP* (4/20, 20%)] and fosfomycin [*fosA* (18/20, 90%)] resistance in *P. aeruginosa*. None of these ARGs were plasmid-associated, and they were not found in clusters on the chromosome. In *P. saudiphocaensis* and *P. stutzeri*, the carbapenem-resistant gene *blaPST-2* was present, but we did not find any carbapenem-resistant genes in carbapenemase negative *Pseudomonas* species. In *P. aeruginosa*, an average of 225 virulence genes were identified per isolate, whereas other *Pseudomonas* spp. lacked virulence genes (Table 2). Virulence genes were associated with biofilm formation, secretion system, pili, and fimbriae formation.

Five carbapenemase positive *Acinetobacter* spp. isolates were sequenced, with three isolates being identified as *A. haemolyticus* and the other two as *A. lwoffii* and *A. junii*. At least one carbapenem-resistant gene was identified in all *Acinetobacter* spp. including *blaOXA-265* (*A. haemolyticus*, *n* = 3), *blaOXA-648* (*A. lwoffii*, *n* = 1) and *blaOXA-278* (*A. junii*, *n* = 1). In *A. haemolyticus*, the aminoglycoside (*aacA-ACI*) and cephalosporin (*blaPDC-197*) resistance genes were also identified (Table 1).

In carbapenemase positive *S. maltophilia*, genes associated with carbapenem (*blaL1*, *blaL2*), aminoglycosides (*aph(6)-Smalt*, *aph(3′)-IIc*), and chloramphenicol (*floR2*) were identified (Table 1). We also identified a multidrug efflux RND transporter operon (*oqxB9*, *oqxA10*) responsible for quinolones resistance. This multidrug efflux RND transporter was mapped twice on *S. maltophilia* genome along with *aph(3′)-IIc*. As expected, we did not identify any carbapenemase-associated ARGs in *O. intermedium*, as this isolate was negative for carbapenemase production. However, ARGs conferring resistance to chloramphenicol (*floR*), quinolones (*oqxB12*) and cephalosporin (*blaOCH-2*) were identified in this isolate.

Hybrid genome assemblies allowed complete circular genomes to be constructed for some of the isolates (Appendix A). Moreover, some ARGs, including *blaOXA-265* in *A. haemolyticus*, and *blaPST-2* in *P. saudiphocaensis*, were only identified after hybrid assemblies (data not shown) were generated.

### 2.4. Comparative Genomic Analysis of Pseudomonas aeruginosa Isolates from Bovine and Human Clinical Origin

Core-genome phylogenomic analysis formed two clades where the majority of *P. aeruginosa* recovered from bovine sources clustered together in a paraphyletic clade (Figure 2). In human clinical isolates, collectively, we identified 20 different ARGs conferring resistance to aminoglycosides [*aac(3)-Ic*, *aac(3)-Id*, *aac(3)-VIa*, *aac(6′)-Ib4*, *aac(6′)-Ib-AKT*, *aac(6′)-Ib-G*, *aac(6′)-IIa*, *aac(6′)-IIc*, *aac(6′)-Il*, *aacA8*, *aadA1*, *aadA5*, *aadA6*, *ant(2″)-Ia*, *ant(4′)-IIb*, *aph(3′)-Ib*, *aph(3″)-Ib*, *aph(3′)-IIb*, *aph(3′)-VIa*, and *aph(6)-Id*], while in bovine *P. aeruginosa* only 3 genes, *aph(3″)-la*, *aph(3′)-IIb*, and *aph(6)-ld* were present (Appendix A).

All clinical isolates carried *blaOXA-50* variants (Figure 3). In addition, metallo-carbapenemase gene *blaNDM-1* was found in two clinical genomes (2/75; 3%), but not in any of the bovine isolates. We also identified ARGs for cephalosporins *(blaCTX-M-30*, *blaGES-9*, *blaOXA-10*, *blaOXA-101*, *blaOXA-2*, *blaPDC-15*, *blaPDC-16*, *blaPDC-22*, *blaPDC-34*, *blaPDC-38*, *blaPDC-46*, *blaPDC-55*, *blaPDC-59*, *blaPDC-100*, *blaPDC-101*, *blaPDC-116*, *blaPDC-123*, *blaPDC-167*, *blaPDC-264*, *blaPDC-308*, *blaPDC-364*, *blaPDC-374*, *blaPDC-385*, and *blaVEB-9*), chloramphenicol (*cmlA5*, *catB*, *catB7*, *catB*, and *floR2*), colistin (*mcr-1*), quinolones (*crpP* and *qnrVC1*), fosfomycin (*fosA*), macrolides [*mcr(E)* and *mph(E)*], sulfonamide (*sul1*), tetracycline *[tet(G)*] and trimethoprim (*dfrA1*, *dfrA10*, and *dfrB5*) in human clinical *P. aeruginosa* genomes. Comparison of *blaOXA-50* gene variants and their genomic context did not reveal any significant variation across genomes from human and bovine sources (Figure 3).

## 3. Discussion

Increasing carbapenem resistance has threatened the clinical utility of these drugs in human medicine, leading to the challenge of “extreme drug resistant” bacteria [22]. Therefore, the Centers for Disease Control and Prevention (CDC) has declared carbapenem-resistant Enterobacteriaceae as an urgent and serious threat to human health [23]. In this study, we investigated if carbapenem-resistant *E. coli* could be recovered using carbapenem enrichment from bovine feces or catch basin water samples collected in intensive feedlots. We had extremely low recovery of carbapenem-resistant *E. coli* in bovine feces. The majority of isolates that were presumed to be *E. coli* were subsequently identified as other bacterial species, primarily *Pseudomonas* species. This is not surprising as we used etrapenem for sample enrichment, and *Pseudomonas* spp. are intrinsically resistant to this antimicrobial [24,25].

In *Acinetobacter* spp., the carbapenem resistance genes identified were mostly intrinsic [26]. For instance, *blaOXA-214*, *blaOXA-278*, and *blaOXA-648* in *A. haemolyticus*, *A. junii*, and *A. lwoffii* isolates, respectively, are intrinsic to these species [27,28]. *A. lwoffii* is a medically important *Acinetobacter* species, however, *A. haemolyticus* and *A. junii* have been only occasionally reported in human infections [29]. The *blaL1* and *blaL2* genes encoding for metallo-β-lactamase in *S. maltophilia* are also intrinsic and confer broad spectrum resistance against β-lactams including all carbapenems [30]. *S. maltophilia* is considered a “newly emerging pathogen of concern” and is frequently isolated from immunocompromised patients [31,32,33].

In *P. aeruginosa*, we identified four different gene variants belonging to *the blaOXA-50* class (*blaOXA-50*, *blaOXA-902*, *blaOXA-486*, and *blaOXA-494*) encoding carbapenem-hydrolyzing oxacillinase (CHDLs). All of these variants were also present in clinical isolates in addition to seven other variants *blaOXA-395*, *blaOXA-488*, *blaOXA-847*, *blaOXA-848 blaOXA-901*, *blaOXA-905*, and *blaOXA-906*. It has been reported that *blaOXA-50* naturally exists in all *P. aeruginosa* and does not appear to have been acquired based on the similar GC% content of the *blaOXA-50* gene to the overall *P. aeruginosa* genome [24]. We did not find any insertion sequences, repeat elements, relaxases, integrases, within the genomic proximity of *blaOXA-50*, suggesting that this gene lacks mobility and is unlikely to be readily transferred to other bacterial species.

In the core-genome phylogenetic analysis, *P. aeruginosa* isolates from bovine sources clustered separately from those obtained from humans. Similar observations have been reported in Brazil, where *P. aeruginosa* recovered from healthy bovine urine samples segregated from human clinical urinary tract isolates [34]. These observations might be associated with a prolonged association and adaptation of these isolates within their respective environmental hosts. *P. aeruginosa* is known for its remarkable ability to adapt to diverse ecological niches, from soil to various living hosts [35,36]. Genes associated with metabolism and pathogenesis constitute the core-genome of *P. aeruginosa*, whereas genes required to adapt to various niches constitute the accessory genome. These genes were found to cluster in certain loci referred to as ‘regions of genomic plasticity’ [37,38]. In this study, *P. aeruginosa* recovered from the beef production system carried ARGs conferring resistant to chloramphenicol (*catB7*), β-lactams (*blaOXA-50*), fosfomycin (*fosA*), aminoglycosides (*aph(3′)-Iib*), and cephalosporins (*blaPDC-55*, *blaPDC-374*). These ARGs were also found in the genomes of human clinical isolates and were mapped on chromosomes in both bovine and human clinical isolates. In *P. aeruginosa*, the type IV pilus uptakes foreign DNA during transformation events [39,40]. The accessory genome can be distinguished from the core genome by its aberrant GC content, codon usage, and tetranucleotide usage [41]. The GC content of *P. aeruginosa* (~66.3%) is mostly higher than the foreign DNA. Over time, the acquired DNA may lose the sequence compositional difference that distinguishes it from the core genome of *P. aeruginosa* as it undergoes the same selective pressure as the core genome [42].

We observed variable phenotypic susceptibility profiles among *P. aeruginosa* isolates that harbored similar AMR profiles. Some isolates were resistant to all carbapenems, some were resistant to both meropenem and doripenem and some were only resistant to meropenem. Other than β-lactamase production, alterations or lack of porin OprD, and overexpression of resistance-nodulation-division (RND) efflux pumps (MexAB-OprM, MexCD-OprJ, MexEF-OprN, and MexXY-OprM) are also associated with carbapenem resistance in Pseudomonas species [43,44]. Not only does the combinations of these mechanisms confer reduced susceptibility to carbapenems, but the overexpression of these efflux pumps result in β-lactam, tetracycline, trimethoprim, aminoglycosides, and fluoroquinolone resistance [45,46]. It is more likely that the variation in susceptibility profiles in our isolates is associated with the varied number of *MexAB-OprM* operon and *OprD* found in each isolate [47,48]. Moreover, isolates with multiple *MexAB-OprM* operons also possessed a single copy of the *MexR* repressor gene. The presence of only one repressor gene in comparison to multiple efflux pumps as regulatory targets may disturb the molecular stoichiometry of the regulation affecting efflux pump expression, thus resulting in varied phenotypic resistance profiles among *P. aeruginosa* isolates.

*P. stutzeri* (*Stutzerimonas stutzeri*) is an opportunistic pathogen that rarely causes infection in humans [49]. In our study, we found the carbapenem-resistant gene, *blaPST-2* in the *P. stutzeri* genome. The *blaPST-2* encodes for a subclass B1 metallo-β-lactamase and was first identified on the chromosomes of *P. stutzeri* DSM 10701 [50]. From a genetic context, we did not find genes associated with mobility in proximity to this ARG. However, we suggest that *blaPST-2* is not intrinsic to *P. stutzeri* as it was not present in all *P. stutzeri* genomes in the NCBI database (only 4 out of 19 *P. stutzeri* genomes had *blaPST-2*). Moreover, *blaPST-2* was phylogenetically related to the previously characterized mobile subclass B1 metallo-β-lactamase families, including KHM, SIM, and IMP in *P. aeruginosa* and *Citrobacter freundii*, providing further evidence that this gene was more likely acquired [51,52,53].

*Pseudomonas saudiphocaensis* has only recently been classified as a new species [54] and has not been well characterized. There are only two studies where *P. saudiphocaensis* was recovered, first from air samples in the city environment of Makkah, Saudi Arabia, in 2012 [54] and then from a sheep dairy farm in New Zealand [55]. To the best of our knowledge, this is the first report of recovery of *P. saudiphocaensis* from bovine feces. With the hybrid genome assembly, we were able to construct a complete circular genome of this isolate. The genome of *P. saudiphocaensis* (3.6 Mbp) was small as compared to other *Pseudomonas* species, which ranged from 5.5 to 6.7 Mbp. These isolates also carried *blaPST-2*, which has not been identified in other *P. saudiphocaensis* genomes [54,55]. The *blaPST-1* in *P. stutzeri* and *P. saudiphocaensis* showed 95% amino acid similarity. Despite limited knowledge of *P. saudiphocaensis*, it is suggested that it may have acquired this gene.

## 4. Materials and Methods

### 4.1. Sampling, Isolation and Identification

The study includes bovine fecal and catch-basin water samples, collected from four feedlots (pens = 301) located in Alberta, over a period of two years (August 2016 to June 2019). Briefly, 20 g of feces were collected from 20 fresh fecal pats and placed in a sterile plastic container and thoroughly mixed. Each pen housed approximately 180 feedlot cattle. For enrichment, 0.5 g of the mixed fecal sample was inoculated into 4.5 mL of *Escherichia coli* (EC) broth containing 2 µg/mL of ertapenem (ETP, sigma Aldrich, ref: sml1238), followed by overnight incubation at 37 °C in a shaking incubator at 250 RPM. The enriched samples were then sub-cultured on MacConkey agar supplemented with 2 µg/mL ertapenem. From each sample, a maximum of three colonies were selected and sub-cultured on nutrient agar supplemented with 0.5 µg/mL ertapenem (Dalynn Biologicals, Calgary, AB, Canada). Water samples were collected from feedlot catch basins which collected run off water from the feedlot. Briefly, 1 L of catch-basin water was collected into a polyethylene bottle attached to a telescopic pole at two different locations per site, which were combined to generate a composite sample. Water samples were transported on ice within 4 h of collection for processing. Composite catch basin water (10 mL) was filtered through a 0.45 µm pore size filter of 47 mm diameter (S-Pak^®^ EMD Millipore Corp, Billerica, MA, USA) using a sterile vacuum-manifold filtration system (Pall Corporation, Port Washington, NY, USA). The filter was then submerged in 4.5 mL EC broth-ertapenem (0.5 mg/L) and incubated overnight at 37 °C with shaking, followed by sub-culture onto MacConkey Agar supplemented with 0.5 µg/L ertapenem at 37 °C (Dalynn Biologicals, Calgary, AB, Canada). A maximum of three colonies were selected from each sample and sub-cultured on nutrient agar supplemented with 0.5 µg/mL ertapenem at 37 °C for 24 h. For the identification of recovered isolates (*n* = 116), the 16S rRNA gene was amplified using universal bacterial 16S rRNA gene primers 27F (5′-AGAGTTTGATCCTGGCTCAG-3′) and 1492R (5′-GGTTACCTTGTTACGACTT-3′) followed by Sanger sequencing of the amplified PCR product [56]. Species were identified using BLAST search against the NCBI bacterial database.

### 4.2. Phenotypic Characterization

Isolates (*n* = 116) were tested against 12 different antimicrobials including all four carbapenems (ertapenem, meropenem, doripenem, and imipenem), ceftazidime, chloramphenicol, gentamicin, levofloxacin, piperacillin, trimethoprim-sulfamethoxazole, tobramycin, and tetracycline, using the disk diffusion method according to the Clinical and Laboratory Standards Institute (CLSI) guideline M02-A12 and M100-S32. *Pseudomonas aeruginosa* ATCC 27853 was used as a reference quality control. Zones of inhibition were recorded using the BioMic V3 imaging system (Giles Scientific, Inc., Santa Barbara, CA, USA).

Isolates were also tested for the production of carbapenemases using the chromogenic Carba NP test (RAPIDEC^®^ CARBA NP kit, BioMérieux, St-Laurent, QC, Canada), with *Klebsiella pneumoniae* ATCC 700603 and *Klebsiella pneumoniae* OLC2685 used as negative and positive controls, respectively. A bacterial colony (10-μL loop) was picked up from overnight-incubated Mueller-Hinton agar plates and mixed into API suspension medium. The bacterial suspension was then transferred to wells in a test strip and incubated at 37 °C. Test strips were read at 30 and 120 min. A ‘positive’ test corresponded to a color change from red to yellow-orange, while no-change in color was considered ‘negative’.

### 4.3. Whole-Genome Sequencing, Assembly and Annotation

All Carba NP positive isolates (*n* = 28) and a subset (*n* = 14) of Carba NP negative isolates selected to be representative of identified species were subjected to whole genome sequencing using short and long reads sequencing technologies. High molecular weight genomic (HMW) DNA was extracted using a Genomic DNA preparation kit with Genomic-tip 20/G (Cat: 13323; QIAGEN, Germantown, MD, USA) according to manufacturer’s instructions. DNA quality and quantity was estimated using a Nanodrop 2000 spectrophotometer (Thermo Fisher Scientific, Mississauga, ON, Canada) and a Qubit Fluorometer with PicoGreen (Q32850, Invitrogen, Carlsbad, CA, USA), respectively. The integrity of DNA was confirmed through agarose gel electrophoresis. Short-read sequencing was performed on the Illumina MiSeq platform. A genomic library was constructed using the Illumina NexteraXT DNA sample preparation kit (Illumina Inc., San Diego, CA, USA) followed by sequencing on Illumina MiSeq platform using the MiSeq Reagent Kit V3, generating 2 × 300 base paired-end reads. Long-read sequencing was performed on PromethION platform from Oxford Nanopore technologies (ONT). Sequencing libraries were prepared using LSK-109 genomic DNA preparation kit. One hundred nanograms of HMW genomic DNA were end-repaired using a NEBNext Ultra II End prep enzyme mix [Cat: E7646AA; New England Biolabs (NEB) Ltd. Whitby, ON, Canada] and FFPE DNA repair mix [Cat: M6630L; New England Biolabs (NEB) Ltd. Whitby, ON, Canada]. End-repaired DNA from each isolate were individually barcoded using a Nanopore barcoding kit EXP-NBD196 [Oxford Nanopore Technologies (ONT), UK] and a blunt TA ligase [Cat: M0367; (NEB) Ltd. Whitby, ON, Canada], followed by chelation with 0.5 M pH 8.0 EDTA (Cat: AM9260G; Invitrogen, ON, Canada). Barcoded DNA samples were pooled in a single tube and cleaned using the Omega-bind NGS beads (M1378-01, Omegabiotek) at 0.5× volume following the manufacturer’s instructions. Beads were dried for 1 min and DNA was eluted in 32 µL of deionized sterile H_2_O. The DNA concentration was measured using the Invitrogen Qubit dsDNA BR assay (Q32854, Invitrogen, ON, Canada). Sequencing adapters were then ligated to 30 µL of recovered DNA using Adapter Mix II Expansion kit EXP-AMII001 (ONT) along with the 5 µL T4 DNA ligase [Cat: M0202M; New England Biolabs (NEB) Ltd. Whitby, ON, Canada] and 10 µL quick ligation buffer [Cat: B6058S; New England Biolabs (NEB) Ltd. Whitby, ON, Canada]. Adaptor-ligated DNA was bead-cleaned using a 0.4× volume of beads following the manufacturer’s instructions using the supplied polyethylene glycol (PEG) based wash buffer. The DNA from the last step (200–400 ng) was loaded onto an ONT PromethION sequencing flow cell as directed by the manufacturer. MinKNOW Core 3.1.20 and guppy 2.0.10 were used for flow cell signal processing and base calling during each run, and reads were assembled de novo using Flye version 2.9.1 [57].

Hybrid genome assemblies were generated using illumina short-reads and Flye-assembled contigs from ONT long-reads using the Unicycler assembly tool [58]. Illumina paired-end short-read sequences were used to generate an assembly graph using SPAdes followed by bridge building using Flye-assembled contigs using Miniasm (https://github.com/lh3/miniasm accessed on 29 January 2023) [59] and Racon (https://github.com/isovic/racon accessed on 29 January 2023). Multiple rounds of short-read polishing were conducted using Pilon (https://github.com/broadinstitute/pilon accessed on 29 January 2023). The contiguity and quality of each genome assembly was assessed using Quast (v.5.2.0) by computing relevant metrics, including the number of contigs, total length (bp), and GC content [60]. Contigs were then annotated using Prokka v.1.13.1 [61], and assembled contigs were screened for the presence of antimicrobial resistance and virulence genes using ABRicate v.1.0.1 (https://github.com/tseemann/ABRICATE accessed on 10 February 2023) against the NCBI Bacterial Antimicrobial Resistance Reference Gene Database (NCBI BioProject ID: PRJNA313047) and the VirulenceFinder database (PMID: 34850947) [62], respectively. To identify plasmids, Mob-recon tool v.3.0.0 (https://github.com/phac-nml/mob-suite accessed on 12 Febrary 2023) was used [63].

### 4.4. Comparative Genomic Analysis

Whole genome comparative genomic analysis was conducted between all the *Pseudomonas aeruginosa* isolates (*n* = 20) sequenced in this study to *P. aeruginosa* genomes (*n* = 76) (Appendix A) originating from North America, from the PathoSystems Resource Integration Center (PATRIC) (https://www.patricbrc.org; accessed on 20 February 2023). Although these same genomes were also available from the NCBI database, the metadata in the PATRIC database was more detailed. These isolates originated from human clinical sources, with the exception of a single isolate from cattle.

The core-genome phylogenomic tree was constructed using the SNVphyl pipeline v.1.2.3. The phylogenetic tree was generated by aligning paired-end reads against the *P. aeruginosa* POA1 reference genome (NC_002516.2) using SMALT (c.0.7.5; https://sourceforge.net/projects/smalt/ accessed on 10 March 2023). The generated read pileups were then subjected to quality filtering (minimum mean mapping quality score of 30), coverage cut-offs (15× minimum depth of coverage), and a single nucleotide variant (SNV) abundance ratio filter of 0.75 to obtain a multiple sequence alignment of SNV-containing sites. The SNV alignment, with no density filtering, was used to create a maximum likelihood phylogeny using PhyML version 3.0. The generated Newick file was visualized using the Interactive Tree Of Life (iTOL) v.6 [64].

We also compared variants of the *blaOXA-50* of *P. aeruginosa* isolates used in this study to determine their genetic relatedness. For this, *blaOXA-50* coding sequencing (CDS) was computationally extracted from each genome and aligned using MAFFT version 7.490. The resultant alignment-based tree was visualized using iTOL v.6 [64].

## 5. Conclusions

In conclusion, the recovery of carbapenem-resistant bacteria from beef production systems is quite rare. The majority of carbapenem-resistant bacterial species, including *P. aeruginosa*, *A. haemolyticus*, *A. junii*, *A. lwoffii*, and *S. maltophilia*, carried intrinsic carbapenem-resistant genes and were only recoverable following enrichment. *P. aeruginosa* found in this study were multidrug-resistant, a possible reflection of the ability of this species to readily acquire foreign genes via transformation through the type IV pilus. Moreover, the phylogenetic analysis showed that bovine *P. aeruginosa* strains formed separate clusters from human clinical strains, indicating that they may have adapted to the cattle environment, and prudent management in feedlot systems has limited the spread of these isolates outside of cattle production systems.

## Figures and Tables

**Figure 1 antibiotics-12-00960-f001:**
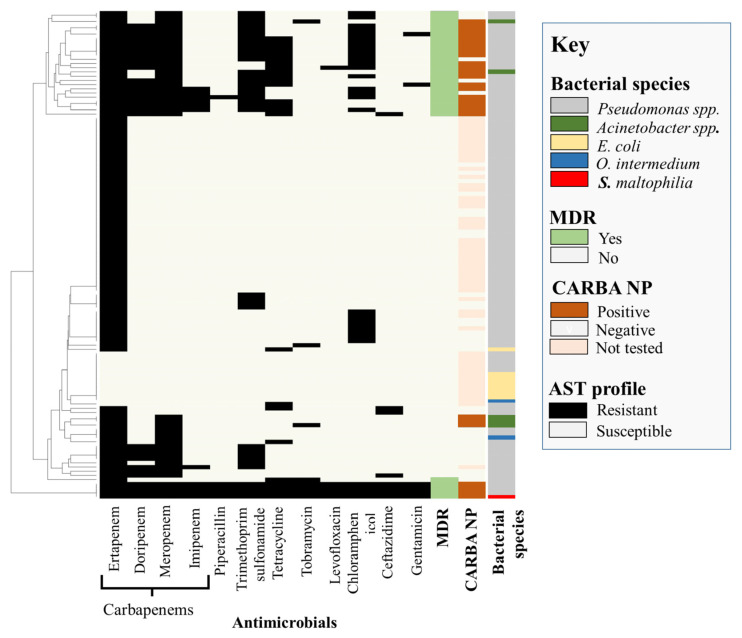
Antimicrobial susceptibility and carbapenemase production phenotypes of isolates recovered from cattle feedlots following carbapenem enrichment.

**Figure 2 antibiotics-12-00960-f002:**
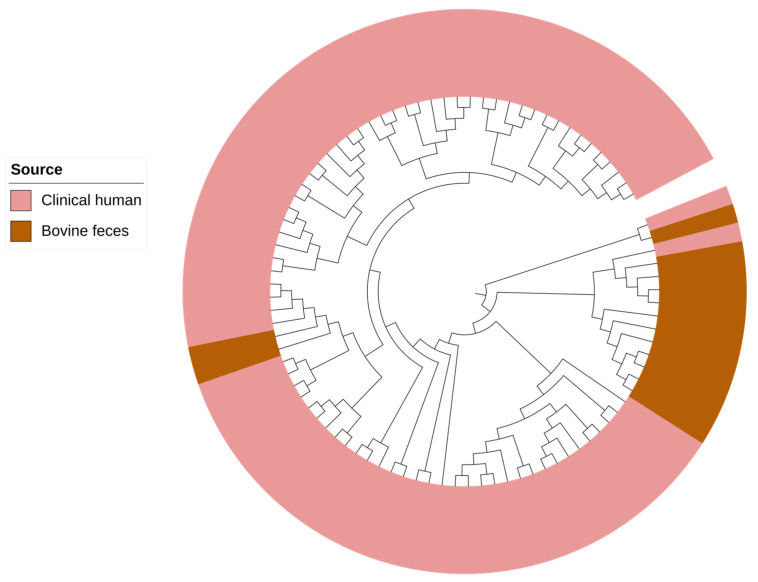
Maximum likelihood core-genome phylogenetic tree of *Pseudomonas aeruginosa* recovered from bovine and human clinical sources. *P. aeruginosa* POA1 (NC_002516.2) was used as a reference genome.

**Figure 3 antibiotics-12-00960-f003:**
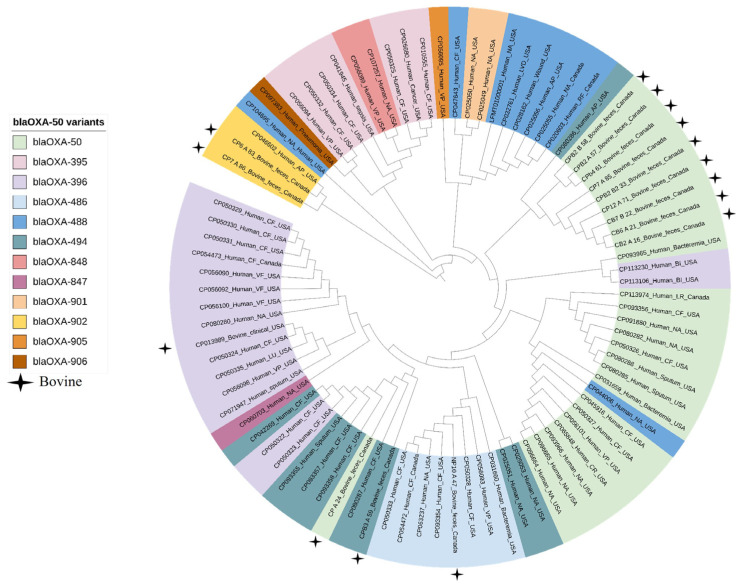
Phylogenetic tree of carbapenemase gene *blaOXA-50* variants identified in *Pseudomonas aeruginosa* isolates from human and bovine sources, constructed using MAFFT.

**Table 1 antibiotics-12-00960-t001:** Antimicrobial resistance gene profiles of whole genome sequenced isolates recovered from bovine feces and catch basin water samples.

Bacterial Species	Antimicrobial Resistance Genes	Genotype-Based Phenotype
*E. coli* (*n* = 1)	*blaEC, blaCMY-2, aph(3″)-Ib, aph(6)-Id, sul2,* *tet(A), floR*	Aminoglycosides, Chloramphenicol, Sulfisoxazole, Tetracycline
*A. haemolyticus* (*n* = 3)	*blaOXA-265, aacA-ACI, blaPDC-197*	Carbapenem Aminoglycosides Cephalosporin
	*blaOXA-265, aacA-ACI1*	Carbapenem Aminoglycosides
*A. lwoffii * (*n* = 1)	*blaOXA-648*	Carbapenem
*A. junii* (*n* = 1)	*blaOXA -278*	Carbapenem
*S. maltophilia* (*n* = 1)	*blaL1,blaL2, aph(6)-Smalt, aph(3′)-IIc, oqxB9, oqxA10, floR2*	Carbapenem Aminoglycosides Phenicol Chloramphenicol
*O. intermedium* (*n* = 1)	*floR, oqxB12, blaOCH-2*	Chloramphenicol Quinolone Cephalosporin
*P. aeruginosa* (*n* = 20)	*blaOXA-50, blaPDC-197, aph(3′)-IIb, catB7, fosA*	Carbapenem Cephalosporin Chloramphenicol Aminoglycoside Fosfomycin
	*blaOXA-50, blaPDC-197, aph(3′)-IIb, catB7, fosA, crpP*	
	*blaOXA-50, blaPDC-55, aph(3′)-IIb, catB7, fosA*	
	*blaOXA-50, blaPDC-66, aph(3′)-IIb, fosA, crpP, catB7*	
	*blaOXA-486, blaPDC-374, aph(3′)-IIb, fosA, crpP, catB7*	
	*blaOXA-486, blaPDC-374, aph(3′)-IIb, catB7, fosA*	
	*blaOXA-494, blaPDC-374, aph(3″)-la, aph(6)-ld, catB7 fosA, crpP*	
	*blaOXA-902, blaPDC-133, aph(3′)-IIb, catB7, fosA, crpP*	
	*blaOXA-902*	Carbapenem
*P. entomophila* (*n* = 1)	*blaPDC-33*	Cephalosporin
*P. plecoglossicida* (*n* = 9)	No gene	-
*P. mosselii* (*n* = 1)	No gene	-
*P. putida* (*n* = 1)	No gene	-
*P. saudiphocaensis* (*n* = 1)	*blaPST-2, aadA1*	Carbapenem, Aminoglycoside
*P. stutzeri* (*n* = 1)	*blaPST-2*	Carbapenems

**Table 2 antibiotics-12-00960-t002:** Virulence gene profiles of whole genome sequenced isolates recovered from bovine feces and catch-basin water samples.

Bacterial Species	Virulence Genes (%) ^1^
*P. aeruginosa* (*n* = 20)	**Biofilm and capsule synthesis:** *alg44* (21/24, 88%), *alg8* (22/24, 92%), *algA-G,I-L,P-R,U,W,X,Z* (22/24, 92%), *mucA-E,P* (21/24, 88%) **Pili and Fimbriae:** *chpA*-E (92%), *pilA* (9%), *pilB* (88%), *pilC* (34%), *pilE,F* (84%), *pilG-K,M-X,Y1,Y2* (92%), *fimT-V* (84%), *fleI/flag* (13%), *fleN* (100%), *fleP* (13%), *fleQ* (100%), *fleR* (88%), *fleS* (88%), *flgA* (84%), *flgB* (88%), *flgC* (100%), *flgD-F* (84%), *flgG-I* (24), *flgJ,K* (84%), *flgL* (13%), *flgM,N* (84%), *flhA,B,F* (100%), *fliA,C* (100%), *fliD* (13%), *fliE-L* (92%), *fliM,N* (100%), *fliO-R* (88%), *fliS* (13%), *motA-D,Y* (92%) **Biosynthesis of small ferric-ion-chelating molecules:** *pchA-I,R* (84%), *fptA* (84%), *fpvA* (21%), *mbtH-like* (100%), *pvcA-D* (84%), *pvdA* (84%), *pvdD,E* (21%), *pvdF-H* (88%), *pvdI,J* (21%), *pvdL,M* (100), *pvdM* (22), *pvdN-Q* (88%), *pvdS* (96%) **Phenazine biosynthesis:** *phzA1-G1,M,S* (80%), *phzH* (30%) **Rhamnolipids:** *rhlA,B,I* (84%), *rhlC* (75%) **Type VI secretion system:** *clpV1* (96%), *dotU1* (88%), *flha1* (80%), *hcp1* (96%), *hsiA1* (84%), *hsiB1/vipA* (100%), *hsiC1/vipB* (96%), *hsiE1,F1,H1* (84%), *hsiG1* (96%), *hsiJ1* (92%), *icmF1/tssM1* (84%), *tagQ* (88%), *tse1-3* (84%), *vgrG1a* (80%), *vgrG1b* (84%), *ppkA*(84%), *tagR* (100%) *tagS* (88%), *tagT* (88%), *pppA* (84%), *tagF/pppB* (84%) **Type III secretion system:** *pscB-L, N-U* (80%), *popB,D,N* (80%), *pcr1-4,D,H,R,V* (80%), *exsA*-E (80%), *exoS,T,Y* (80%), *ptxR* (84%) **Type I secretion system:** *aprA* (84%) **Type II secretion system (Xcp) and exo-proteins:** *xcpP-Z* (84%), *xcpA/pilD* (88%), *lasA* (88%), *lasB* (84%), *plcH* (84%), *toxA* (21%), *lip1* (84%) **Quorum sensing:** *lasI* (88%) **Lipopolysaccharide core biosynthesis:** *waaA,C* (88%), *waaF* (96%), *waaG,P* (100%), *wzy* (17%), *wzz* (17%)
*P. entomophila* (*n* = 1)	**Biofilm and capsule synthesis:** *algCBU, mucD* **Lipopolysaccharide core biosynthesis:** *waaF, wag* **Type VI secretion system:** *clpV1, hsiG1, hcp1, hsiC1/vipB, hsiB1/vipA, tagR* **Biosynthesis of small ferric-ion-chelating molecules:** *pvdH, pvdS*, *mbtH-like* **Pili and Fimbriae:*** motC, fleNQ, flhA, fliAIGMNPQ, flgCGHI, pilH*
*P. mosselii* (*n* = 1)	**Biofilm and capsule synthesis:** *algA-D,U,W,I* (100%), *alg8* (100%)*, mucD* (100%) **Biosynthesis of small ferric-ion-chelating molecules:** *mbtH-like* (100%)*, pvdH,S,M* (100%) **Lipopolysaccharide core biosynthesis:** *waaF,G,P* (100%) **Pili and Fimbriae:** *flgC,G-I* (100%)*, fliA,F,G,I,M,-Q* (100%)*, fleN,Q* (100%)*, motA-C* (100%)*, pilH* (100%) **Type VI secretion system:** *tagR, dotU1* (100%)*, hsiB1/vipA* (100%)*, hsiC1/vipB* (100%)*, hcp1*(100%)*, hsiG1* (100%)*, clpV1* (100%)
*P. putida* (*n* = 2)	**Biofilm and capsule synthesis:** *algA-D,U,I* (100%),*alg8* (100%), *mucD* (100%) **Biosynthesis of small ferric-ion-chelating molecules:** *mbtH-like* (100%)*, pvdH,S* (100%) **Lipopolysaccharide core biosynthesis:** *waaF,G* (100%) **Pili and Fimbriae:** *flgC,G-I* (100%)*, fliA,G,I,M,N,P,Q* (100%)*, fleN,Q* (100%)*, flhA* (100%)*, motC,D* (100%)*, pilH* (100%)
*P. saudiphocaensis* (*n* = 1)	**Biofilm and capsule synthesis:** *algC,R,U* (100%) **Pili and Fimbriae**: *flgC,G,I* (100%)*, flhA* (100%)*, fliE,G,I,M-P* (100%), *pilG,H,U,T* (100%) **Type II secretion system (Xcp):** *xcpT,R* (100%) **Lipopolysaccharide core biosynthesis:** *waaF*
*P. stutzeri* (*n* = 1)	**Biofilm and capsule synthesis:** *algA-C,R* (100%) **Pili and Fimbriae**: *flgG,I* (100%)*, flhA* (100%), *fliA,E-G,I,M,N-R* (100%)*, fleN,Q* (100%)*, motA* (100%)*, pilG,H,J,M,R,T,U* (100%) **Type II secretion system (Xcp):** *xcpT,R* (100%) **Lipopolysaccharide core biosynthesis:** *waaF,P* (100%)
*P. plecoglossicida* (*n* = 9)	**Biofilm and capsule synthesis:** *algB,C,U* (100%), *algW* (67%), *mucD* (100%) **Pili and Fimbriae:** *flgC,G,H,I* (100%)*, flhA* (100%)*, fliA,G,I,M,N,P,Q* (100%), *fleN,Q* (100%)*, motC* (100%), *pilH* (100%) **Lipopolysaccharide core biosynthesis:** *waaF* (67%)*, waaG* (100%) **Type VI secretion system:** *clpV1* (100%), *hcp1* (100%), *hsiB1/vipA* (100%), *hsiC1/vipB* (100%), *hsiG1* (100%), *tagR* (100%) **Biosynthesis of small ferric-ion-chelating molecules:** *mbtH-like* (100%), *pvdSH* (100%)
*E. coli* (*n* = 1)	**Type II secretion system:** *gspC m* (100%) **Type III secretion system**: *espX2* (100%)*, ompA* (100%)*, espR1*(100%)*, espR4* (100%)*, espR3* (100%)*, espL1* (100%)*, espY3* (100%)*, espY2* (100%)*, espX1*(100%)*, espY4* (100%)*, espL4* (100%)*, espX4* (100%)*, espX5* (100%)*, * **Curli biogenesis:** *csgB,D,F,G* (100%) **Iron import system:** *shuA,S,T,W,X* (100%)*, chuUVW* (100%) **Pili and Fimbriae:** *fimA-H* (100%)*, yagV/ecpE* (100%)*, yagW/ecpD* (100%)*, yagX/ecpC* (100%)*, yagY/ecpB, yagZ/ecpA* (100%)*, ykgK/ecpR* (100%) **Adhesion:** *fdeC* (100%) **Enterobactin:** *entA-E,F,S* (100%)*, fepA-D,G* (100%)
*A. haemolyticus* (*n* = 3)	No virulence gene
*A. lwoffii * (*n* = 1)	
*A. junii* (*n* = 1)	
*S. maltophilia* (*n* = 1)	

^1^ Bold indicates virulence genes associated with cellular component or metabolic activity of each specific bacterial species.

## Data Availability

The whole genome sequence data are available in GenBank under Bio Project PRJNA956966.

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
