# Peer review of "Genomic Characterization of Carbapenem-Resistant Bacteria from Beef Cattle Feedlots"

_antibiotics, 2023, doi:10.3390/antibiotics12060960_

Round 1

Reviewer 1 Report

Review of 'Genomic characterization of carbapenem resistant bacteria from beef cattle feedlots'

The authors report the distribution of antibiotic resistant bacteria from cattle feedlots. In addition to reporting on the antibiotic resistant profiles of the several different bacterial species recovered, they also carried out genomic sequencing of selected isolates and identified virulence and antibiotic resistance elements in those genomes. This is a straightforward paper that should be of interest to those working in the area of antibiotic resistance.

I have some small points that need attention:

1: The introduction should describe (briefly) the various carbapenem antibiotics that are discussed later on in the MS.

2: There are problems with all the figures; in the pdf file that I was able to download, all of the figures are unacceptable fuzzy, to the extent that some parts are illegible. Fix this.

3: Figure 1: explain what the phylogenetic tree on the left hand side is about - this should be referred to in the figure legend.

4: Figure 2 has 2 panels, each with separate legends, but both referred to simply as 'Figure 2'. Either make 2 separate figures, or differentiate them as Fig 2a and Fig 2b, with a single legend.

5: There are numerous examples where the authors have been careless about italicizing genus and species names. This is particularly evident in lines 110-120. Again, go through the entire MS carefully and fix this.

6. Table 1: What , if any, is the difference or significance between n=01 vs. n=1, in the 'Bacterial species' column? Use one or the other, but be consistent.

7. line 141: Were any of the AGRs linked, or clustered in the chromosome?

8: Lines 237-239; re-write this, combining the two sentences and correcting the grammatical errors.

9: lines 266-268; replace 'might have acquire this gene two reasons.' with 'might have acquire this gene for two reasons.'  Also, the subsequent 2 sentences really don't make much sense and I can't figure out what your point is, or how these sentences explain acquisition of the resistance gene.

10: lines 33-39 - remove these instructions to authors.

11: I was unable to access the supplemental materials.

see above

Author Response

Reviewer 1

1: The introduction should describe (briefly) the various carbapenem antibiotics that are discussed later on in the MS.

Required information has been provided as requested in introduction

Line 45-48 ‘So far, four carbapenems including ertapenem, meropenem, doripenem and imipenem have been approved for use in the US. These members differ in their side chains, influencing their antimicrobial activity.’

Line 53-57 ‘Ertapenem binds preferentially to PBPs 2 and 3 of Escherichia coli and has low affinity for PBP 1a, 1b, 4 and 5. Imipenem binds with high affinity to PBP2, followed by PBP1a and 1b, but binds to PBP3 with low affinity. Meropenem possesses high affinity for PBP 2, 3 and 4. Doripenem, similar to meropenem can bind to PBPs 2 and 3 of P. aeruginosa and also has affinity for PBP2 of E. coli [2].’

2: There are problems with all the figures; in the pdf file that I was able to download, all of the figures are unacceptable fuzzy, to the extent that some parts are illegible. Fix this.

Figure quality has been improved. Font size of legend and key is increased and now is more readable. All figures are in 600dpi. The pdf converter reduces the quality of figures, therefore high resolution figures in JPEG format are provided separately.

3: Figure 1: explain what the phylogenetic tree on the left hand side is about - this should be referred to in the figure legend.

It is an association tree not a phylogenetic tree. It represents the association of isolates based on their phenotypes

4: Figure 2 has 2 panels, each with separate legends, but both referred to simply as 'Figure 2'. Either make 2 separate figures, or differentiate them as Fig 2a and Fig 2b, with a single legend.

Figure 2 is a phylogenetic relatedness tree based on whole genome sequence data, where each source (human and bovine) has been colored differently, it does not represent two separate panels. This tree was created to identify if human and bovine isolates are phylogenetically related to each other or not. Creating two separate trees for human and bovine isolates will not provide us any information about their relatedness.

5: There are numerous examples where the authors have been careless about italicizing genus and species names. This is particularly evident in lines 110-120. Again, go through the entire MS carefully and fix this.

Corrected

  1. Table 1: What , if any, is the difference or significance between n=01 vs. n=1, in the 'Bacterial species' column? Use one or the other, but be consistent.

Corrected in Table 1.

  1. line 141: Were any of the AGRs linked, or clustered in the chromosome?

No, none of the ARGs were linked or clustered on the chromosomes (line 150).

8: Lines 237-239; re-write this, combining the two sentences and correcting the grammatical errors.

Rephrased and corrected

9: lines 266-268; replace 'might have acquire this gene two reasons.' with 'might have acquire this gene for two reasons.'  Also, the subsequent 2 sentences really don't make much sense and I can't figure out what your point is, or how these sentences explain acquisition of the resistance gene.

Further clarification has been added: However, it is suggested that blaPST-2 is not intrinsic to P. stutzeri as this gene was not present in all P. stutzeri genomes found in the NCBI database (only 4 out of 19 P. stutzeri genomes had blaPST-2). Moreover, blaPST-2 was phylogenetically related to the previously characterized mobile subclass B1 metallo-β-lactamase families, including KHM, SIM, and IMP in P. aeruginosa and Citrobacter freundii, providing further evidence that  that this gene was more likely acquired[60-62].

10: lines 33-39 - remove these instructions to authors.

Removed

11: I was unable to access the supplemental materials.

Supplementary tables are resubmitted.

Reviewer 2 Report

In this work, the occurrence of carbapenem resistant bacteria in feedlots in Alberta were investigated. 27% of the isolates (n=31) were resistant to at least one other carbapenem. 90% were carbapenemase positive. Whole genome sequencing identified intrinsic carbapenem resistance genes. However, some changes could improve the manuscript.

Specific Comments:

1.       Figure 1 is blurred. Additionally, authors should explain the right panel in figure legends.

Furthermore comments:

1.       The written needs strengthen. There are servaral mistakes.

2.        Line33-39, the authors would better delete them.

 The written needs strengthen. There are servaral mistakes.

Author Response

Reviewer 2

  1. Figure 1 is blurred. Additionally, authors should explain the right panel in figure legends.

Figure quality is improved. Font size of legend and key is increased and now is more readable. All figures are in 600dpi. The pdf converter has reduced the quality of figures. High resolution figures in JPEG format are provided separately.

Furthermore comments:

  1. The written needs strengthen. There are several mistakes.

Following changes has been made to improve the quality of manuscript

  • Extra spacing has been removed in the manuscript (line 44, 68, 99
  • Species name was italicized in line 90, 120, 174
  • Correction has been made in line 45
  • Sentence has been rephrased in line ‘we investigated genomic relatedness among Pseudomonas aeruginosa isolates recovered in this study to previously published genomes through comparative analysis.’
  • Brief introduction to carbapenems members has been provided in the introduction section.

Line 45-48 ‘So far, four carbapenem including ertapenem, meropenem, doripenem and imipenem have been approved for use in the US. These members differ in their side chains, influencing the antimicrobial activity.’ Line 53-57 ‘Ertapenem binds preferentially to PBPs 2 and 3 of Escherichia coli and has low affinity for PBP 1a, 1b, 4 and 5. Imipenem binds with high affinity to PBP2, followed by PBP1a and 1b, but to PBP3 with low affinity. Meropenem possesses high affinity for PBP 2, 3 and 4. Doripenem, similar to meropenem can binds to PBPs 2 and 3 of P. aeruginosa and also has affinity for PBP2 of E. coli [2].’

  • In Discussion section, lines 278-283 has been rewritten for more clarification ‘However it is suggested that blaPST-2 is not intrinsic to stutzeri as this gene was not present in all P. stutzeri genomes found in the NCBI database (only 4 out of 19 P. stutzeri genomes had blaPST-2). Moreover, blaPST-2 was phylogenetically related to the previously characterized mobile subclass B1 metallo-β-lactamase families, including KHM, SIM, and IMP in P. aeruginosa and Citrobacter freundii, indicating that this gene has more likely been acquired as well [60-62]..
  1. Line33-39, the authors would better delete them.

These lines have been deleted

Reviewer 3 Report

In this manuscript, the authors characterize the genotype and phenotype of carbapenem resistant bacteria recovered from beef cattle production feedlots after carbapenem enrichment.

Line 44: Format extra space

Line 45: Sulfone in, incomplete sentence

Line 61: Format extra space

Line 75, 76: Needs rephrasing. Genomic analysis/ genomic relatedness sounds redundant

Line 81: Italicize Pseudomonas

Line 90:  Edit as "was also found to be resistant"

Line 99: Italicize Pseudomonas

Line 99: "production" redundant

Quality of Figure 1 is really low. Needs clear legend and key.

Line 112: Italicize Pseudomonas

Line 164: Italicize haemolyticus

Line 172: Which species are referred here as human clinical isolates?

Line 213: Format extra space

Line 224: Format punctuation in the end

Author Response

Reviewer 3

Line 44: Format extra space

Corrected

Line 45: Sulfone in, incomplete sentence

Corrected

Line 61: Format extra space

Extra space is removed

Line 75, 76: Needs rephrasing. Genomic analysis/ genomic relatedness sounds redundant

Sentence has been rephrased in line 75 (now 82-86) ‘we investigated genomic relatedness among Pseudomonas aeruginosa isolates recovered in this study to previously published genomes through comparative analysis.’

Line 81: Italicize Pseudomonas

Corrected

Line 90:  Edit as "was also found to be resistant"

Corrected as suggested

Line 99: Italicize Pseudomonas

Corrected

Line 99: "production" redundant

Correction has been made on line 99

Quality of Figure 1 is really low. Needs clear legend and key.

Figure quality is improved. Font size of legend and key is increased and now is more readable. All figures are in 600dpi. The pdf converter has reduced the quality of figures. High resolution figures in JPEG format are provided separately.

Line 112: Italicize Pseudomonas

Corrected

Line 164: Italicize haemolyticus

Corrected

Line 172: Which species are referred here as human clinical isolates?

Modified sub heading to Comparative genomic analysis of Pseudomonas aeruginosa isolates

Line 213: Format extra space

Extra space has been removed

Line 224: Format punctuation in the end

Corrected

Round 2

Reviewer 2 Report

This manuscript can be accepted.

none